# Fear, Efficacy, and Environmental Health Risk Reporting: Complex Responses to Water Quality Test Results in Low-Income Communities

**DOI:** 10.3390/ijerph19010597

**Published:** 2022-01-05

**Authors:** Saskia Nowicki, Salome A. Bukachi, Sonia F. Hoque, Jacob Katuva, Mercy M. Musyoka, Mary M. Sammy, Martin Mwaniki, Dalmas O. Omia, Faith Wambua, Katrina J. Charles

**Affiliations:** 1School of Geography and the Environment, University of Oxford, South Parks Road, Oxford OX1 3QY, UK; sonia.hoque@ouce.ox.ac.uk (S.F.H.); jacob@fundifix.co.ke (J.K.); katrina.charles@ouce.ox.ac.uk (K.J.C.); 2Institute of Anthropology, Gender and African Studies, University of Nairobi, IAS Museum Hill, Parklands Highridge, Nairobi 00100, Kenya; salome.bukachi@uonbi.ac.ke (S.A.B.); mercymbithe93@gmail.com (M.M.M.); dalmas.ochieng@gmail.com (D.O.O.); faithwambua00@gmail.com (F.W.); 3FundiFix Miambani Ltd., Mbithe Kimotho Building, Ngaie-Tseikuru Road Junction, Kyuso Centre, Kitui County 90200, Kenya; mary.sammy@fundifix.co.ke (M.M.S.); martin.mwaniki@fundifix.co.ke (M.M.)

**Keywords:** environmental health, behaviour change, intervention development, risk communication, drinking-water safety, rural water services

## Abstract

Reducing disease from unsafe drinking-water is a key environmental health objective in rural Sub-Saharan Africa, where water management is largely community-based. The effectiveness of environmental health risk reporting to motivate sustained behaviour change is contested but as efforts to increase rural drinking-water monitoring proceed, it is timely to ask how water quality information feedback can improve water safety management. Using cross-sectional (1457 households) and longitudinal (167 participants) surveys, semi-structured interviews (73 participants), and water quality monitoring (79 sites), we assess water safety perceptions and evaluate an information intervention through which *Escherichia coli* monitoring results were shared with water managers over a 1.5-year period in rural Kitui County, Kenya. We integrate the extended parallel process model and the precaution adoption process model to frame risk information processing and stages of behaviour change. We highlight that responses to risk communications are determined by the specificity, framing, and repetition of messaging and the self-efficacy of information recipients. Poverty threatscapes and gender norms hinder behaviour change, particularly at the household-level; however, test results can motivate supply-level managers to implement hazard control measures—with effectiveness and sustainability dependent on infrastructure, training, and ongoing resourcing. Our results have implications for rural development efforts and environmental risk reporting in low-income settings.

## 1. Introduction

Risk communications that describe a hazard with the purpose of motivating behaviour change are often conceptualized as fear appeals [1]. They are used extensively for public health messaging and the fear appeals literature, which has developed over 7 decades, offers varied views on the persuasiveness and optimal design of appeals on a range of topics including the management of domestic and wider environmental exposures [1]. Access to information impacts perceptions of environmental health risks, including from drinking-water [2], and information-based feedback loops are key determinants of behaviour in complex adaptive systems [3]. Research has evaluated the impact of risk communications on diverse issues, from preventing contraction of malaria [4] or human immunodeficiency virus (HIV) [5] to reducing neurotoxin-producing cyanobacterial blooms in water bodies [6]. Here, we focus on the impact of sharing drinking-water quality test results in low-income rural communities. We recognize that in order to have sustained impact, interventions that promote uptake of water safety measures must engage with systemic complexity [7,8,9,10].

In rural Sub-Saharan Africa, where water access is largely reliant on community managed supplies or self-supply [11,12], water-related disease is an important public health challenge [13,14,15]. Since the 1980s, international and national development policies have focused on infrastructure design and quality of construction as a means to improve the sustainability and safety of supplies while devolving responsibility for operations and maintenance to the community level [16,17]. The legacy of these policies continues to shape rural water supply, with many project designs and timelines underpinned by “Western ‘cultural idealization’ of communities in low-income countries” [18] (p. 366). Poor performance of the community-based management model, with widespread operational failure rates between 30 to 60%, has been attributed to ineffective institution building [18,19]. In response, evolving risk logics are shifting focus from discrete infrastructure projects towards ongoing service provision through increasingly pluralistic institutional arrangements—recognizing that communities need ongoing support to effectively manage water supplies [17,20,21]. There are established arguments for ‘going with the grain’ so that rural water project approaches are sensitive to power, accountability, and social morality norms [22] and, on the other hand, research has highlighted that institutional development must challenge entrenched hierarchies, which maintain inequalities and distributive injustice [23,24,25].

The literature on institutional change in the rural water sector has focused on maintaining functionality of improved infrastructure, with water safety largely a secondary consideration [24,26]. Research on the quality of water supplied by improved infrastructure demonstrates that these supplies are widely susceptible to microbial and chemical contamination [27,28]. Consequently, international norms and standards now call for monitoring and active management of water quality to be included in rural water service delivery, as reflected in the target tracking framework for Sustainable Development Goal 6.1 [11]. Practice has been slow to reflect this policy shift [29,30], but some governments, regulators, and donors are looking at options to increase rural water safety monitoring [31,32,33].

As the sector grapples with institutional questions of how and by whom monitoring should be conducted and reported [34], it is important to consider how results may serve near-term operational decision-making in addition to regulatory or policy-making purposes [35]. Research from Ethiopia and Kenya showed that infrastructure monitoring focused on functionality can increase rural water supply resilience by informing operation and maintenance activities [36,37], but operational decision-making for water safety was not addressed. Recognizing that operational water management responsibilities in rural Sub-Saharan Africa continue to rest primarily with communities, we ask whether, and under what conditions, water quality monitoring could be useful for improving management at two key levels: the supply-level and the household-level. At supply-level, the caretakers of rural water supplies are often lay water managers (LWMs) who have neither professional qualifications nor access to water quality testing equipment. At household-level, water users influence the safety of their drinking-water by their choice of supplies and use of household water treatment (HHWT) and safe storage practices [38].

Studies have shown that uptake and sustained use of water safety measures is complex [7,9,10]. Additionally, evidence on the impacts of disseminating water quality test results on water safety behaviours is ambiguous, with studies focusing on microbial contamination and/or arsenic and reporting variable findings [39,40,41,42,43,44,45,46,47]. A 2011 review concluded that “rigorous studies on this topic are needed” [45] (p. 8) and since then studies in a range of contexts (e.g., from Bangladesh [39,40], Cambodia [41], Ghana [47], India [43,46], Tanzania [42], and the USA [44]) have continued to report variable findings. The literature points to the complexity of the relationship between data reporting and behavioural responses and highlights a need for research with a greater temporal scope to understand long-term behaviour change (and the impact of repeated messaging). It also indicates that contextual factors (such as intracommunity variability in socioeconomic conditions, prior knowledge of water safety, perceptions of information trustworthiness, and competing priorities) are important and under-studied. Most research in this space has taken a quantitative approach, but the value of qualitative inquiry to contextualize results has been demonstrated [48]. Additionally, studies have focused on water safety at the point of use (household-level) without examining the potential for LWMs to be change agents for water safety at the supply-level, despite examples of engagement at this level advancing water safety outcomes [49].

In this study, we conceptualize water quality results-reporting as a form of fear appeal and we investigate the potential effects of sharing water quality monitoring data with rural water users and LWMs. Our objectives are to (a) assess user perceptions of drinking-water quality hazards and (b) evaluate an information intervention through which microbial water quality monitoring results were shared with LWMs over a 1.5-year period. In the materials and methods section, we explain our integrated fear appeal framework and our use of mixed-methods to layer insights from water quality testing, cross-sectional and longitudinal surveys, and semi-structured interviews. In the results section, we present our household-level and LWM analyses separately. In the discussion section, we integrate our findings and reflect on the implications for the design and evaluation of environmental risk reporting in low-income settings.

## 2. Materials and Methods

### 2.1. An Integrated Fear Appeal Framework

Recognizing that sharing water quality results could induce negative affective responses in individuals where water is unsafe, we find that conceptualizing monitoring reports as fear appeals provides a useful framing. Most fear appeals research has evaluated short-term (<2 weeks) outcomes of a single message [1], so there is limited theory and empirical evidence of sustained behaviour change or the influence of repeated messaging [50]. Consequently, we place the fear appeal process in the wider structure of a stage change model [51,52], employing a hybrid framework that incorporates both drivers and stages of behaviour change (Figure 1). Our framing focuses on decision-making by individuals, but acknowledges that daily water management decisions are complex and embedded in household and community systems [53]. It directs our assessment to focus on cognitive and affective message processing while accounting for time and individual and situational differences.

The fear appeals literature offers a rich theoretical foundation for framing the cognitive and affective processing of health risk information. In particular, the extended parallel process model (EPPM) [54], which consolidates concepts from protection motivation theory [56], the parallel process model [57], and the fear-as-acquired drive model [58], has been influential for health campaign design [50] and research in diverse areas including communication, health policy, psychology, business, and information security. It posits that an external stimulus causing an increase in perceived threat and consequent negative affective state (fear) can motivate changes in beliefs and behaviours.

The change manifests either through problem-focused processing—when efficacy is high—or defensive processing—to reduce fear and/or cognitive dissonance when efficacy is low. Defensive processing refers to cognitive responses such as avoidance, denial, reactance, suppression, and re-appraisal, and is based in research on emotional regulation [59]. Problem-focused processing refers to development of beliefs, intentions, and behaviours that engage with and mitigate the threat itself as opposed to the negative emotion that arises from being confronted with it. The central concepts of threat and efficacy each have two parts: threat has dimensions of susceptibility (the likelihood of experiencing a threat) and severity (the magnitude of the consequences of a threat) and efficacy relates to both response efficacy (the effectiveness of a measure for controlling a threat) and self-efficacy (one’s personal ability to take measures to control a threat). The model also includes an ‘individual differences’ parameter where contextual considerations are grouped.

The EPPM is widely considered to be conceptually strong but lacking in reliable and consistent operational definitions [60]. Empirical studies that have operationalized the model in mathematical terms, often without explicit consideration of temporal effects or context, have had inconsistent results [61,62]. Consequently, we have used the core concepts of the EPPM to guide our study design and thematic analysis, but we do not use the model in a predictive capacity and we avoid operationalizing the concepts of threat and efficacy as continuous numeric variables. Furthermore, to better account for the influence of time and baseline context, we draw on a second, complementary behaviour change model: the precaution adoption process model (PAPM) [63]. In contrast to the EPPM, the PAPM uses stages of change to conceptualize behaviour change as a process, making time and precedent experience key considerations in understanding how individuals respond to information about threats.

The PAPM was developed as a theoretical framework for evaluating the influence of risk messages on adoption of household radon tests [63]. It is similar to the widely used transtheoretical stages of change model [64], but is more appropriate for our study context because, having been developed with reference to reducing harmful environmental exposures, it is not concerned with addiction (e.g., to unhealthy behaviours such as smoking) and recognizes ‘unaware’ as a distinct stage. The PAPM posits that when individuals receive general information about a threat, they move from being unaware (stage 1) to being aware but uninvolved in considering precaution measures (stage 2). Upon receiving information about the threat that is specific to themselves, they are prompted to consider adopting precautionary measures (stage 3, which aligns with message processing in the EPPM). If a decision is made, the stages then split with individuals either having decided not to act (stage 4, which aligns with a defensive response in the EPPM) or having developed an intention to act (stage 5, which aligns with a problem-focused response in the EPPM). For precautions to be adopted, individuals must act (stage 6) and, when relevant, maintain the adopted behaviour (stage 7).

The drivers of transitions between stages are not fully established as core concepts in the PAPM, but those that are well-supported [55] are included in Figure 1. An increase in general knowledge (driver A: general external stimuli) shifts individuals from stage 1 to 2. Increasing specificity about the relevance of the threat to an individual (driver B: specific external stimuli) prompts stage 3. The outcome of stage 3 is driven by the interaction between perceived threat and efficacy, affective state, and individual difference and, as framed by the EPPM, results in no response (driver C: insufficient threat perceived), defensive response (driver D: defensive motivation), or problem-focused response (driver E: protection motivation). Decision-making in stage 3 may be extended by driver F (information seeking), which is a precursor to developing intentions around action [65]. Neither the EPPM nor the PAPM provide well-established drivers at stages 6 and 7, for which habit formation and actual, as opposed to perceived, efficacy are expected to be important [55]. Additionally, low information recall, defensive response outcomes, and failed problem-focused responses are logically expected to cause reversion to stages 1, 2, or 3 as indicated by the dotted lines in Figure 1.

By integrating the EPPM with the PAPM, we frame behaviour change in response to a fear appeal as a process while retaining theoretical depth for conceptualizing message processing. The integrated framework is advantageous because it distinguishes between (a) individuals who are ‘unaware of’ versus ‘uninvolved’ with a threat; (b) appeals that provide ‘general’ versus ‘personally specific’ information; and (c) information seeking, intentions to act, action, and maintenance of action.

### 2.2. Study Area

The empirical work for this study was conducted in Mwingi North, the northernmost sub-county of Kitui County, Kenya (Figure 2). Kitui County is the sixth largest in Kenya (30,430 km^2^) with 1.1 million residents, 95% of whom live in rural areas [66] where agropastoral livelihoods are predominant and households are vulnerable to droughts, floods, and food price fluctuations [67]. Access to water infrastructure is limited: 39% of residents report that their main drinking source is surface water [67]. This is worse than average for Kenya where 52% of the rural population have access to basic water (an ‘improved’ supply located less than a 30 min round-trip away including queuing time) and 24% collect surface water [68]. Diarrheal disease is estimated to account for 7.8% of disability adjusted life years (DALYs) in Kitui County, 6.5% of DALYs in Kenya, and 8.2% of DALYs in Sub-Saharan Africa, with 70–90% of diarrheal disease attributed to unsafe water supplies [13]. Access to safe water has been a constitutional right in Kenya since 2010, and the 2016 Water Act established that County governments are responsible for rural water provision [32]. The Kenyan water services regulator and the Kitui County government express interest in developing monitoring programs to support rural water safety management [33,34].

### 2.3. Mixed-Methods Data Collection

To assess perceptions of drinking-water safety and evaluate LWM responses to microbial water quality monitoring results, we collected data between March 2018 and December 2020 using 6 instruments (Table 1). The first three capture user perceptions. We started with a cross-sectional household survey on indicators of multidimensional poverty and domestic water practices (Table 1 instrument 1). We selected households from Mwingi North sub-county by mapping villages on a 10 km grid and randomly sampling 40 households around each village (Figure 2). A subset of these households was then selected for the longitudinal work (Table 1 instrument 2): the sampling frame was restricted to two wards for logistical efficiency and households were grouped by a 3 × 2 matrix of main water supply concern (costly, unsafe, other) and upper or lower wealth quartiles. Wealth quartiles were determined using an index calculated by principal component analysis of 25 variables as per [69]. Households were randomly selected within each grouping and invited to attend 2-h training sessions in groups of 20. Five households dropped-out after training and 115 participated in the study. Additionally, 35 water users from 2 communities were interviewed to qualitatively explore perceptions of water safety (Table 1 instrument 3).

We decided against sampling and sharing water quality results at household-level due to concerns about low self-efficacy. This was based on discussions with government stakeholders [34] and on the results of the cross-sectional survey. We reasoned, however, that LWM responses to water quality monitoring results warranted further investigation. By designing a monitoring programme in collaboration with a local water maintenance service provider, we could ensure ongoing informational support and explore the possibility of establishing a supply chain for treatment resources or training if LWMs expressed interest. In keeping with our integrated fear appeal framework and to understand the influence of test result variability on attitudes and behaviour, we prioritized repeated measures over cross-sectional sample size. We selected 52 LWMs to represent different management arrangements—including community-based management (CBM) committee members, school or health facility administrators, and private owners—and different water supply types (extraction and distribution setups). Handpumps (12), piped schemes (25 sourcing groundwater from boreholes and 4 sourcing surface water from reservoirs), earth dams (3), and open wells (5) were included. We focused more on handpumps and piped supplies rather than earth dams and open wells, which have less active management and less variable water quality.

In late 2018, we established a fit-for-purpose laboratory in Kyuso, Kitui, as the base of operations for the water quality monitoring (Table 1 instrument 4). The lab is equipped with customized work benches and shelving, a fridge and freezer to store consumables and ice-packs, two small incubators for the bacteriological work, and a solar-charged battery back-up system to ensure consistent power supply. Water points were sampled monthly and *Escherichia coli* results were reported to the LWMs. Chemistry results were not shared in keeping with directives from the County Ministry of Water following concerns about the limited self-efficacy of LWMs to manage salinity and fluoride, and potential political implications. Consequently, while the chemistry results provide useful context to understand the multiple water quality threats that must be managed in the study area, we were not able to assess LWM responses to water chemistry information.

Reporting of the *E. coli* results was conducted in Kiswahili, Kikamba, or English according to the LWM’s preference. The first reporting was conducted in-person; a hard-copy information sheet was used as a guide for explaining the results and was shared with each contact for future reference (Appendix A). It included information explaining the World Health Organization *E. coli* risk categories and water safety response options [70] and was created collaboratively (with input from 3 LWMs and 5 Mwingi North residents who had prior experience discussing water use with respondents during the cross-sectional household survey). Subsequent reporting was done verbally over the phone to minimize the time between completing the analysis and sharing the result.

We used a series of surveys to explore the LWMs’ reactions to the monitoring information and track changes in their perceptions, intentions, and behaviours in keeping with our integrated fear appeal framework (Table 1 instrument 5). Additionally, we conducted semi-structured interviews to gain richer insight into LWM perceptions of water safety and the utility and drawbacks of the monitoring programme (Table 1 instrument 6). The questions encouraged LWMs to reflect on their experiences of water quality, the monitoring results, options for managing water safety, and stakeholder roles and responsibilities for responding to monitoring information.

**Table 1 ijerph-19-00597-t001:** Data collection for the water user perceptions assessment (instruments 1 to 3) and the LWM information intervention evaluation (instruments 4 to 6).

Instrument	Dates	Sample	Description	Execution
(1) Cross-sectional household survey	9–20 March 2018	A total of 1457 households: 71% household heads, 22% spouses, 6% other relatives. 60% were between 30 and 59 years old and 44% presented as female.	A survey on indicators of multidimensional poverty, including domestic water services, with a subsection on perceptions and decision-making around drinking-water safety. The questionnaire, further method information, and data are available via the UK Data Service ReShare public repository (10.5255/UKDA-SN-854561).	17 enumerators were locally recruited and trained on data collection, ethics, and code of conduct. Tablets and the Open Data Kit and Enketto mobile survey platforms were used. Quality control checks were executed daily with continuous feedback to enumerators. The final data were checked for consistency and coherence, with incomplete forms excluded.
(2) Longitudinal household survey (daily water diaries and bimonthly check-in survey)	August 2018–July 2019	One hundred fifteen households: completed 5826 diary weeks and 1241 check-in surveys (min 4, max 19, mean 11 surveys each). A total of 78% of participants presented as female and mean age was 45 (SD = 15 years).	Daily diary forms and twice-monthly surveys to capture participants water collection practices. Details of the forms and their development are already published [71]. The twice-monthly surveys asked about perceived changes in water quality, HHWT practices, and illnesses (see questionnaire in Appendix A).	Three enumerators conducted the check-in surveys and collected and digitized the diary forms using tablets and the Ona Systems survey platform. Participants did a pilot week to practice. Data were reviewed bi-monthly by 2 of the authors. Follow-up visits with participants to seek explanations for unusual data were conducted by the enumerators as needed. Mid-week check-in phone calls were implemented to counteract disinterest.
(3) Household-level semi-structured interviews and participant observation	July–November 2018	Thirty-five water users: primary fetchers of water (13), and/or primary managers of water within the home (17), and/or household heads (19). A total of 18 presented as female, 6 were single parenting, 5 were physically disabled, and 4 were >60 years old.	Qualitative depth sourced from 35 semi-structured in-depth interviews and 21 participant observation daily journals, exploring the diversity of water perceptions and practices within communities, including which factors influence supply-selection. Further information about the ethnographic approach used for this work is published in [72].	Two of the co-authors (female University of Nairobi graduate students) lived in the communities to build rapport and interact with community members at homesteads, water supplies, market areas, and during special functions. The fieldwork began with establishing support from community leaders, familiarizing with the location, and conducting informal scoping conversations and observations to adjust the interview guides. Interviews were conducted in Kiswahili and Kikamba, and translated and transcribed in English.
(4) Water quality monitoring programme	December 2018 –2020	Seventy-nine water points: 12 handpumps, 52 piped groundwater taps from 25 schemes (including 10 mixed tanks with rainwater collection), 3 earth dams, 5 open wells, and 7 piped surface water taps from 4 schemes.	Monthly sampling and analysis of chemical (pH, conductivity, turbidity, fluoride) and microbial (*Escherichia coli* and total coliforms) water quality in 2019. Parameters were selected following Kitui water supply project document review and meetings with the County government to identify key contaminants of concern for the area. In 2020, we continued quarterly sampling for 45 of the sites (those which were registered for maintenance services).	Two of the co-authors conducted the monitoring with training and supervision from the first author. On-site testing was conducted using a HACH multimeter (HQ 40D) with a conductivity (CDC40101) probe, and a Hanna turbidimeter (HI93703). pH (PHC10101), fluoride (ISEF12101), and *E. coli* and total coliforms (IDEXX Quanti-Tray2000 system with Colilert-18 growth medium) were analysed in the lab. Details of the sampling protocol including weekly quality assurance and control steps are in Appendix A.
(5) LWM survey series	November 2018–July 2020	Fifty-two LWMs: 28 CBM committee members, 15 school administrators, 3 health facility officers, and 6 private owners. 81% presented as male. The median level of participation in the survey series was 86% (limited primarily by supply functionality and periodic absence of LWMs).	A series of 5 surveys to track LWMs’ reactions to the monitoring results including changes in their perceptions, intentions, and behaviours around drinking-water safety. The surveys were conducted (1) before monitoring, (2) after the first reporting, (3) monthly check-ins, (4) at the end of 2019, (5) mid-2020. Appendix A provides an overview of the integrated fear appeal framework concepts assessed in each survey. The questionnaires are provided in Appendix A.	Two of the co-authors conducted the surveys with training from the first author. They lived and worked in rural Kitui prior to this project and each have >4 years of training and experience in water services. Most of the survey questions were open-ended and responses were summarized in paragraphs on paper forms. Each survey was piloted for a week to refine questions and confirm a common understanding of aims among the research team. Data were digitized and checked for comprehensiveness and consistency weekly, with follow-ups for clarification carried out as needed.
(6) LWM semi-structured interviews	July–August 2019	Thirty-eight LWMs: Repeated attempts were made in July and August to interview all 52 LWMs but 4 school and 13 CBM LWMs were not available, and 1 private LWM declined to be interviewed.	Semi-structured interviews to discuss LWM views on water safety; the utility and drawbacks of monitoring; and options, roles, and responsibilities for managing water quality. Conceptual framework terminology was not used in the interview, which was designed to facilitate relatable discussion focused on the practical and specific rather than abstract concepts (see the interview guide in Appendix A).	The interviews were conducted in English (27) or a blend of Kiswahili and Kikamba (11) by the first author and 2 of the co-authors. We used audio recording and verbatim transcription for all but 2 interviews, for which the interviewees preferred that only written notes be used. During post-interview debriefs and the transcription process, ambiguities were discussed by the interview team and annotations were added to guide later analysis. In 4 cases we contacted the LWMs for further clarification.

### 2.4. Data Analysis

Our analysis was guided by the core concepts of our integrated fear appeal framework (Figure 1). For the user perceptions assessment, we used summary statistics, χ^2^ tests, and association plots to interrogate the cross-sectional and longitudinal household survey data. The interview transcripts and participant observation journal entries were coded in two cycles using NVivo 12. Starting deductively with nodes that correspond to perceived threat and efficacy, precautionary actions, and defensive rationale, through the coding process we built more specific nodes to capture key themes that reflect the complexity of water user engagement with microbial water quality threats. We then conducted a second cycle of coding to increase intra-rater consistency. The coding was led by the first author, with feedback on the codebook, coding consistency, and findings from the co-authors who contributed to the design, execution, and transcription of the interviews.

For the LWM messaging intervention evaluation, we drew on the water quality monitoring results, surveys and interviews to track perceptions of threat and efficacy, affect display, information seeking, intended actions, reported actions, and defensive responses. To manage the complexity of the data, we used NVivo 12 for deductive coding and then built a summary timeline for each LWM in Excel. Again, the analysis was led by the first author with feedback at all stages from the co-authors and the coding was conducted in two cycles using nodes corresponding to the integrated fear appeal framework concepts. Using the coded information, we judged stage of change for each LWM across fourteen timeline steps including the end of 2018, twelve months in 2019, and mid-2020.

To assess the associations between LWM perceptions, affect displays, and stages of change, we used correspondence analysis (CA) with contribution bi-plots [73]. CA is an extension of principal component analysis that is useful for categorical variables. It enables exploration of the associations between categories of one variable and categories of another variable in a two-way contingency table. Contribution biplots are a method of visualizing the output of CA in low-dimensional space, where the relative positions of the categories reflect their associations in the contingency table [74]. This analysis was done using the ‘FactoMineR’, ‘vcd’, and ‘factoextra’ packages with R version 3.6.1. Following the CA, we used the LWM timelines to explore the complexity in their responses to the monitoring results. By iteratively grouping the LWMs based on similarities in water quality profiles, perceived efficacy, intended actions, defensive responses, and reported actions, we identified patterns in the LWM responses.

## 3. Results

We present our mixed-method results in three parts: First, we summarize the water quality monitoring results to provide context for the subsequent sections (Section 3.1). Second, we present results from the household surveys, focusing on perceptions of threat and efficacy (Section 3.2) and problem-focused and defensive responses to water quality hazards (Section 3.3). Third, we evaluate the effects of reporting water quality monitoring results to LWMs, focusing on baseline awareness (Section 3.4), changes in perceived susceptibility (Section 3.5), and patterns of response (Section 3.6).

### 3.1. Water Quality Results Overview

Most sites exhibited variability in *E. coli* results over the study period (Figure 3). The least protected sites (earth dams and open wells) most consistently had concentrations above 100 MPN/100 mL. The other sites had a larger range: 8% never had *E. coli* detected, 38% had *E. coli* detected less than 50% of the time, and 20% always had *E. coli* detected. Piped surface water supplies were rarely free of *E. coli* and the groundwater supplies had a broad range of *E. coli*, conductivity, and fluoride results—demonstrating that supply type is an inadequate proxy for water safety or acceptability.

Across the monitoring sites, we observe trade-offs between turbidity and microbial contamination versus conductivity and fluoride (Figure 4). The piped surface water, earth dams and open wells had acceptable conductivity and fluoride concentrations but exceeded potable water standards for turbidity and faecal contamination risk as indicated by *E. coli*. Groundwater samples had lower turbidity, but mean conductivity and fluoride concentrations exceeded guidelines at 57% and 44% of sites, respectively. Storage tanks with a mixture of groundwater and rainwater had fewer fluoride guideline exceedances but worse microbial quality. Although we focus here on microbial contamination, the chemistry results speak to important context: some users and LWMs have concerns about fluoride and high salinity (measured by conductivity). Salinity, particularly, constrains response efficacy by reducing the usability of groundwater.

### 3.2. Water User Perceptions of Threat and Efficacy

The household survey respondents judged water safety most frequently on the basis of either general knowledge about pathogens from faecal contamination (*n* = 554) or not attributing any health problems to drinking-water quality (*n* = 559). Their judgements were also influenced by attribution of illness to water quality, organoleptic factors, and advice from experts (Figure 5). General knowledge about risk of teeth damage from contaminated water was localized, with 96% of responses coming from only eight villages (out of a total of 34 villages). Water quality testing found that as a group these villages did not have higher fluoride concentrations in their water supplies and the reason for their heightened concern remains unknown.

More than half of the 1457 household survey respondents recognized that their drinking-water is not always safe (58%). Of those respondents, 13% reported their water is always unsafe but the majority recognized variability and said that their water is rarely safe (43%) or sometimes safe (44%). Only six respondents (<1%) admitted uncertainty and said they do not know if their water is safe. Welfare quartile and level of education did not predict perception of water quality threats, except that respondents from households with no adults having at least primary education (*n* = 136) were less likely to perceive variability in water safety and more likely to say that their water is always safe (χ^2^ = 14.6; *p* < 0.001). Recognition of water quality threat was strongly related to water source. Respondents from households that were mainly using surface water for drinking during the time of the survey were less likely to say that their water is always safe (χ^2^ = 84.7; *p* < 0.001; Appendix A).

Only 10% of the survey respondents said ‘yes’ when asked if they had ever received information about the safety of their drinking-water. Those who did recall receiving information received it from doctors, health officers and community health volunteers (69%), an NGO (13%), a chief or sub-chief (6%), a water service provider (5%), or other (7%). In the water diary households, respondents who reported cases of stomach pain and diarrhoea were often uncertain of the cause, or linked it to waterborne disease, food poisoning, growing teeth, or other causes such as pregnancy complications, malaria, salinity, stress, or ulcers (Appendix A).

The interviews provided richer insight into perceptions of threat from unsafe drinking-water. The perception of surface water being especially unsafe was prominent (Table 2 theme 1). This view was expressed either as common sense or as a result of learning from health facility staff, community health volunteers, and NGOs. Groundwater, in contrast, was viewed as having better microbial quality and was discussed with reference to trade-offs between chemical and microbial quality (Table 2 theme 2). These views are partially supported by our water quality monitoring results, which confirm consistent contamination of surface water and groundwater chemistry challenges, but also highlight widespread microbial contamination of groundwater supplies (Figure 4), which is underrecognized by users. In accordance with the survey findings on low access to water safety information and high uncertainty about the causes of illness, interview participants also highlighted that their appraisal of water safety is influenced by the absence of specific information (Table 2 theme 3). Their views on the severity of water-related disease were expressed in terms of fears for themselves and others (Table 2 theme 4).

Participants also shared views on their ability to respond to water quality threats. Perceived response efficacy was uniformly high, with nobody questioning the existence of effective protection and treatment measures. Self-efficacy, however, was strongly limited by poverty, gender norms, collective action challenges, and rural isolation (Table 2 themes 5–8).

### 3.3. Household-Level Water Management Choices

Despite the self-efficacy limitations discussed in the interviews, some participants did say that they take supply-selection and treatment measures to improve water safety, and this was reflected in the cross-sectional and longitudinal surveys as well. The water diaries showed that households use between one and four water supplies over the year, with the most common being dug wells, temporary open wells in river-beds, earth dams, and piped water kiosks (Figure 6). These preferred supply types present a range of water quality threats (Figure 4). In the interviews, water quality was reported to be a factor in supply selection but distance and cost are key constraints on choice. Interview participants explained supply selection as a result of many interrelated factors (Table 2 theme 9).

Interview participants also discussed intermittent use of water safety measures in response to specific stimuli (Table 2 theme 10). The insight from these discussions is consistent with the survey results. In the cross-sectional survey, 49% of respondents said that they never treat their drinking-water but 19% of respondents said that the water they drank on the day of the survey had been treated, mostly by boiling (74%) and/or adding chlorine-based disinfectant (30%). In the water diaries surveys, the 115 households reported doing water treatment 327 times over the year. Boiling was the most common method, with 61 households (53%) reporting boiling drinking-water in at least one survey.

Only six households reported taking measures consistently (in more than 90% of the surveys in which they participated), 21 households took measures 50 to 90% of the time, 24 households took measures 25 to 50% of the time, 33 households took measures up to 25% of the time, and 31 households never reported doing treatment or buying bottled water. The proportion of reported treatment was not related to wealth index (*p* = 0.4); in open-ended responses, the reasons participants gave for doing treatment or buying bottled water were because:clean drinking-water was not otherwise available (159 responses from 62 households);to avoid previously experienced illness (102 responses from 27 households);to kill germs/bacteria (63 from 31) or visible worms and insects (14 from 11);to protect a sick person (21 from 14), young child (10 from 3), or visitors (6 from 4);because it was advised by a doctor (12 from 6) or NGO (11 from 4);to avoid salinity (2 from 2);or out of habit (1 from 1).

The inconsistency in applying water safety measures reflects the wider threat landscape and self-efficacy constraints that water users navigate. In the household survey, 68% of respondents listed drinking-water services (including quality, quantity, and reliability) in their top three concerns, which also included education (53%), healthcare (32%), agricultural support (28%), transportation and roads (25%), financial services (20%), employment (19%), and electricity (19%) among others. When asked about their top three concerns for water services specifically, 31% of respondents had no concerns, but others said supplies were too far (51%), insufficient in quantity of water (47%), unsafe for drinking (39%), too costly (25%), dirty for domestic use (20%), and unreliable (14%). In the interviews, participants emphasized that in adopting problem-focused responses, they must balance water quality threats with many others (Table 2 theme 11).

According to our integrated fear appeal framework (Figure 1), the combination of substantial perceived threat and limited self-efficacy should result in defensive processing, cognitive responses that help individuals mitigate the negative emotions that arise when they are confronted with a health threat. Defensive responses can take a variety of forms, and they often occur on a sub-conscious level, which makes them difficult to assess. Avoidance and suppression by their nature are most subconscious, and reactance (dismissing a threat because engaging with it would inhibit one’s behavioural freedom) did not feature in the interviews. Interview participants’ reflections on water safety and other threats did, however, demonstrate cognitive re-appraisal (wherein a hazard is acknowledged but additional beliefs frame it as futile to engage with and/or not personally threatening) (Table 2 theme 12). Compared to cognitive reappraisal, denial of water quality threats was uncommon in the interviews, but it was expressed by three participants: one speaking on behalf of himself and the others reflecting on attitudes in their communities more broadly. Denial took the form of dismissing hazards and “just decid[ing] that water is clean” despite contrary observations and learning. One participant also pointed to lack of specific information and difficulty attributing consequences to water quality as denial-enabling.

### 3.4. Lay Water Manager Baseline Awareness

Only 1 LWM said that they had received microbial water quality test information prior to the study (from an NGO that tested once and reported the water was safe). Six others (12%) said that the borehole drillers (contracted by the government) or an NGO tested the water and told them it was either good for drinking (4%) or very saline and should be used for livestock (8%). Most said the water had not been tested (35%), or that they did not know if it had been tested (29%). Others said that they never received any information after the water was tested by researchers (14%), drillers (8%), or an NGO (2%).

In our initial survey, five LWMs (10%) indicated that they were unaware of potential water quality hazards, relying on the assumption that groundwater is safe (this assumption also featured in the results of the household-level baseline assessment). Most LWMs (81%), however, were grouped in the uninvolved stage, saying that they were uncertain about the water quality, recognizing the potential for the water to be unsafe, but not considering taking precautionary measures. Two LWMs were undecided about whether to do something, two were already acting (using chlorine, advising users to boil), and one perceived high threat but had decided against trying to act due to low self-efficacy.

### 3.5. Changes in Perceived Susceptibility

The information about water protection, treatment, and storage that was shared with the LWMs alongside the monitoring results was kept consistent so that perceived efficacy was not intentionally manipulated with each reporting of the monitoring results. Similarly, the severity component of perceived threat was not manipulated since the information on waterborne disease accompanying the results was kept consistent. The second component of perceived threat, susceptibility, was the key message processing variable that was influenced by the reporting of monitoring data. The monitoring reports to the LWMs emphasized that the higher the concentration of *E. coli*, the higher the likelihood of waterborne disease transmission. Additionally, susceptibility with respect to specific water supplies was contingent on use. If the supply was in regular use, perceived susceptibility was coded based on the *E. coli* concentration per 100 mL sample: low (<0), medium (1–10), high (>10). Thus, it is through the susceptibility variable that our *E. coli* monitoring results are linked with the LWM responses.

This method of assessing susceptibility was supported by the LWMs in two important ways. First, the interviews confirmed that trust in the water quality test results was high because those doing the monitoring were perceived as experts who presented the information in a reliable manner without false exaggeration or minimization to serve ulterior motives. For comparison, this same level of trust was not afforded to information provided by fellow community members or government officials except on a case-by-case basis where trusted relationships were established. Second, when fewer people were using a water point, for example, due to seasonal change in availability of alternate options, kiosk closure (caused by difficulty paying attendants or conflict amongst the committee), or school being on break, the LWMs consistently expressed that the microbial quality was not a near-term priority regardless of the test result or intended future use. Thus, perceived susceptibility was coded as low if the supply had limited use such that the LWM spoke of the microbial quality as inconsequential.

The associations between perceived susceptibility, affect display, and stage of change were assessed using correspondence analysis (CA) with contribution biplots. Comparison of perceived susceptibility and affect display found that positive affect is associated with low perceived susceptibility (χ^2^ = 227, *p* < 0.001, dim1: 92%, dim2: 8%). Further analysis found that change in perceived susceptibility (relative to the preceding timeline step) discriminates better between the other affect display categories (Appendix A; χ^2^ = 315, *p* < 0.001, dim1: 76%, dim2: 14%). Negative affect display was associated with increased and sustained high perceived susceptibility. Other strong associations (Pearson residuals >2) were for:positive affect display with sustained low susceptibility;surprise with increased susceptibility;uncertainty with sustained medium susceptibility; anddisinterest and undetermined affect display with sustained high susceptibility.

Comparing affect display with stage of change, intention to act is least associated with affective state, whereas intending no action is associated with disinterest; indecision is associated with uncertain, negative, and surprised affect displays; and the uninvolved stage is associated with positive affect display (Appendix A; χ^2^ = 339, *p* < 0.001, dim1: 69%, dim2: 21%). Comparing change in perceived susceptibility with stage of change directly, the correspondence solution is dominated by the sustained low susceptibility observations which are strongly associated with the uninvolved stage (Figure 7; χ^2^ = 396, *p* < 0.001; dim1: 90%, dim2: 8%). Other substantial associations (with Pearson residuals >2) were between indecision and increase in susceptibility and between intending no action and sustained medium or high susceptibility. Intending to act is inversely associated with sustained low susceptibility.

### 3.6. Evolution of Stages of Change

The first reporting of monitoring results produced the most uniform shift in stage of change across the participants (Appendix A). Of the thirty-one LWMs who perceived medium or high susceptibility to microbial hazards after the first report, twenty-one expressed intention to act, nine sought further information and therefore stayed in the message processing undecided stage, and one continued to intend no action due to low self-efficacy. Of the LWMs who perceived low susceptibility, nine stayed in the uninvolved stage, but two became undecided and ten expressed intention to act based on the information accompanying the results (despite the tests being *E. coli* negative).

With further reporting in the following months, microbial contamination threat and management information was repeated and the LWMs were introduced to the variability of microbial water quality over time, especially in piped schemes (Figure 3 and Figure 4). The LWM response patterns became more complex (Figure 8). Most LWMs (85%) expressed an intention to act at least once during the study period. The choice of different activities was largely guided by the water supply design including the infrastructure and management arrangements that were in place and by the water supply usage patterns including whether the supply served a facility or the general community. The most common intended measures were disinfection-based treatment, tank cleaning, advising HHWT, and fencing, although reported actions were consistently fewer than expressed intentions (Appendix A). Unlike for household level managers, the range of responses under consideration by the LWMs included seeking support from the government (*n* = 6) or NGOs (*n* = 6). This was moderated by level of isolation, with LWMs in facilities or community management committees being more likely to consider it a viable option, especially if they were based nearer to population centres (as opposed to individual owners or committees in more remote locations).

Perceived efficacy was not manipulated with each reporting of monitoring results, but it was commonly found to decrease over time. After the first reporting, 92% of the LWMs expressed confidence that, if necessary, they could execute response measures to resolve microbial water quality threats. Towards the end of the study, however, 87% of the LWMs expressed that an effective response requires external support including training and ongoing resourcing. This was influenced by experience with trying to act on intentions and recognizing challenges in the process that were not fully appreciated initially. Limited access to resources (including financing and local supply chains) and low confidence or know-how in implementing response measures were the main barriers. In response to realizations of limited response and self-efficacy, 30 LWMs indicated defensive processing at least once, including downward comparison, fatalism, resignation, denial, and avoidance (Appendix A).

Defensive processing arose in response to limited self-efficacy, but the LWMs demonstrated that danger and fear control processing are not mutually exclusive. They voiced intentions to take actions that they knew would partly but not fully control the threat. Different appraisals of self-efficacy were applied to different actions, and water supply system design including the infrastructure and institutional structures around each supply moderated response efficacy. For example, one-time measures such as building a fence were more attractive than ongoing measures like routine chlorination, some tanks are easier to clean than others, chlorination was more feasible for supplies with longer stored water residence times or more consistent flow regimes, and *barazas* (community meetings) to promote HHWT and safe storage were associated with high self-efficacy on the part of the organizers but limited response efficacy due to mixed behaviour of community members in following recommendations.

In this way, efficacy was often partial and problem-focused and defensive responses were expressed contemporaneously. For example, one LWM described water treatment (via automatic chlorine dispenser) being implemented for part of their piped network but said that multiple standpipes were not receiving chlorinated water. In their timeline, they demonstrated intention to act, maintenance of action, and then concurrent defensive responses (including downward comparison, resignation, and reframing) with respect to the untreated part of the piped network.

The combination of variable water quality results and experience-based realizations of efficacy constraints contributed to the proportion of LWMs who were undecided about how to respond to water quality threats (Figure 8). LWM reactions to monitoring information over the study period demonstrated six main patterns of response. These are characterized by differences in perceived threat and efficacy, extrapolation of information over time or to other supplies, and extent of defensive processing. Table 3 reports the number of LWM response timelines best described by each pattern. Perceived threat was insufficient to motivate formation of intentions in seven cases. The most common pattern was long-term variable engagement with microbial water quality threat, with LWMs moving between stages in response to changing perceptions of susceptibility and partial efficacy. For 29 LWMs, there was overlap in the patterns, so Table 3 also reports how many times each was observed as a secondary pattern. The patterns with the most frequent secondary expression were: reverting to the uninvolved stage when perceived susceptibility reduced or voicing intentions to act despite *E. coli*-negative results based on extrapolating information in time or to other supplies.

## 4. Discussion

We presented our household-level and LWM findings separately to support clearer links between our methods and results. Here we integrate the findings to discuss key points on threat information specificity and framing (Section 4.1), repeated messaging and maintaining the problem-focused response (Section 4.2) and situating fear appeals in context (Section 4.3). We note that while the LWMs did not systematically have more training than users on water safety threats, they were often in positions of relatively greater self-efficacy due to gender roles and their leadership positions. Most of the LWM participants were male (79%), reflecting the strong gender norms that were evident in the results of the household-level assessment. They also had access to user fees and more scope to apply centralized control measures and to seek external support. Our findings emphasize the importance of specific threat information and proactively-framed repeated messaging. They also encourage consideration of efficacy constraints, particularly in relation to gender norms and complex poverty threatscapes.

### 4.1. Threat Information Specificity and Framing

Baseline general awareness of water quality threat was high among our study participants, but few consistently acted to improve water safety at household-level and most LWMs were uninvolved (not considering adopting water safety precautions). Participants reflected that, in the absence of specific water quality information, attribution of illness to water becomes challenging. Moreover, trade-offs involving water safety and time, monetary expense, effort, and other risks are more difficult to evaluate. Our water quality results show that groundwater microbial contamination threat was underestimated by many study participants. We also found that microbial versus chemical water quality trade-offs, which were discussed by water users and LWMs in interviews, are widespread—with no water supply types reliably providing water that meets standards for *E. coli*, turbidity, conductivity and fluoride.

Upon first receiving *E. coli*-positive test results, the LWMs universally engaged with the threat (none remained uninvolved) and 77% acted at least once to improve water safety. Thus, our results support the expectation of our integrated fear appeal framework that specific threat information (such as a test result) can motivate behaviour change more effectively than general threat information. Health communications that proffer general information about risks without personally specific data are common in research and practice, but we suggest that their effectiveness is likely to be hampered by attribution and trade-off uncertainty (and by optimism bias [76] and the stability of organoleptic and experience-based judgements [77]).

In noting the motivational advantages of specific threat information, we also emphasize the importance of framing test results to encourage proactive risk management. In this study, monthly results-reporting showed benefits from reinforcing consideration of water safety, particularly for LWMs who were inclined to prolonged indecision and information seeking. This is consistent with previous research on persuasion in marketing [59] and water, sanitation, and hygiene (WASH) messaging [78]. However, we caution that the variability of *E. coli* results, which was considerable during our study period especially for piped schemes, led some LWMs to disengage when tests were *E. coli*-negative. This highlights the importance of contextualizing variability in risk-reporting, which should emphasize proactive decision-making by using sanitary inspection information and avoiding overemphasis on any one sample. We note the value of a fear appeal framework that supports longitudinal analysis and accounts for both drivers and stages of behaviour change.

### 4.2. Repeated Messaging and Maintaining the Problem-Focused Response

Repeated messaging initiatives must also recognize that the motivational effects of affective message processing shift over time. Initial results sharing is most associated with rapid affective processing, whereas subsequent sharing is likely to be more associated with conscious affective processing wherein people have more opportunity to influence their emotions through cognitive means [59]. We observed that, in accordance with our fear appeal framework, the effectiveness of repeated messaging strongly depends on the balance between threat and efficacy (Table 3); where efficacy is insufficient, defensive cognitive processing is likely to have increasing influence over time. Low self-efficacy driven by structural constraints is widespread in our study area, and similarly in many lower-income rural regions. Under these conditions, users and LWMs act to improve water safety through limited (as opposed to multi-barrier) controls and/or action at discrete intervals when threat is relatively high (e.g., when someone is already ill). Our temporal analysis shows that, while changes in perceived threat and affective state motivate LWMs to engage with water safety measures, the effect is less productive and reduces over time in the absence of sufficient efficacy.

We find that LWM efficacy is moderated by water supply system design, highlighting the importance of including safety controls in the technical and institutional design of water projects. This begins with verifying that new or refurbished supplies can provide safe water under a range of conditions, with response options in place to respond to hazards. Further, LWMs require ongoing support to operate and maintain water supplies, and this is widely unachievable with local financing alone [17,79]. The disconnect between the constraints on LWMs self-efficacy and the demands of their intended roles is well-established in the literature on community-based water management, although the focus of post construction support has largely been on functionality without explicit consideration of water safety [17,24]. We note that efficacy was overestimated by most LWMs before they had attempted to engage in problem-focused responses and realized, in the process, the extent of the challenges involved. This suggests that many behaviour change studies overestimate the influence of single information interventions by measuring the development of intentions to act over a short duration—usually no more than two weeks in fear appeal evaluations [1] and no more than a month or two in evaluations of responses to water quality test results [39,40,41,42,43,44,45,46,47]. Again, we note the utility of a fear appeal framework and study design that differentiates intention to act from action and tracks changes over time.

### 4.3. Situating Fear Appeals in Context

The implications of individuals navigating multiple threats simultaneously has also not been well addressed in the literature. Fear appeal research has noted interaction effects between multiple perceived threats; for example, in studying the threat of skin cancer, Cho and Salmon also discuss perceived threat to behavioural freedom in the sun [51]. Generally, however, the consequences of complex poverty threatscapes, and how they are shaped by gender inequality, needs to be better considered in risk communication campaign design. Our results indicate that much of the disconnect between water safety threat perception and problem-focused action in low income areas arises from the “everyday complexity of poverty” [80] (p1). People living in poverty constantly navigate high-consequence trade-offs [81] and work with reduced cognitive bandwidth, especially for processing hazards that have chronic or delayed effects [82]. Furthermore, this can apply to other environmental exposures beyond drinking-water safety, for example cooking smoke or pollution from manufacturing activities [83].

As primary household caretakers, women in particular are frequently targeted in behaviour change research and interventions. However, our interview results indicate that where gender-based inequalities in household and community hierarchies present women with additional threats and additionally constrain their self-efficacy, they are especially unlikely to sustain changes in response to fear appeals without concurrent self-efficacy support. Thus, while constraints on resources, know-how, and effective response options are central to the concept of efficacy, the impact of poverty and gender norms on processing and response to fear appeals is broader than these most apparent constraints.

Considerations of poverty and gender-based inequality are at the heart of a dilemma about whether or not potentially distressing information should be shared with people who may not be in a position to protect themselves [34]. On the one hand, access to information is upheld as a human right, and it is further considered to be a core principle of the human rights to water and sanitation such that “information relating to standards, as well as progress towards meeting those standards, [should be] available and accessible” [84] (p30). This view is supported by systems-thinking that emphasizes the importance of information-based feedback loops [3] and by governance theory that argues for balance between political, market, and community enablement processes [85]. In contrast, on the basis of social justice theory [86], Britz 2004, proposed guidelines for reducing information poverty that included acceptance of withholding information if doing so improves the lives of the information-poor [87].

In contemplating notifying LWMs or water users of contamination, how does one weigh psychosocial stress and the burden of responsibility for an intractable problem with the right to information and self-determination? In contexts of poverty, the results of this study discourage sampling at household-level for the purpose of motivating behaviour change. Emphasizing threat without improving efficacy is not expected to motivate substantial behaviour change for improving household-level drinking-water safety. It may even be counter-productive by reducing demand for safely managed water if defensive reasoning is reinforced. While establishing the monitoring programme, we also discovered that sharing data at household-level would have lowered the acceptability of the programme for government and LWMs due to concerns about the self-efficacy of water users. Furthermore, fear appeal messaging based on household-level *E. coli* sampling would be subject to substantial uncertainty in interpretations of health risk [88]. Interventions that focus on behavioural settings [89] and align with ‘One Health’ approaches for improving household environmental conditions are likely to be better investments. We are not suggesting that information be withheld from water users. If testing is done at household-level for other purposes, those conducting the work should consider that access to information can be a dimension of empowerment at household-level [90]. However, results should be shared with sensitivity to efficacy limitations and the threatscapes that household members are navigating, which may mean expectations of promoting behaviour change should be low.

## 5. Conclusions

We conducted this study to understand whether, and under what conditions, rural water quality monitoring is useful for community-level water management, differentiating the activities of household management (by water users) and supply management (by lay water managers). Our analysis, founded on an integrated fear appeal conceptual framework (Figure 1), has generated insight for advancing safety inclusive rural water service delivery models. Our results emphasize the importance of specific threat information, proactive framing of risk, repeated messaging, and accounting for poverty threatscapes and gender norms. Specific information from monitoring can motivate engagement with hazard control measures and LWMs can be effective change-agents for safe rural water supply if their self-efficacy is supported by appropriate infrastructure and institutional arrangements. In addition to training, rural water supplies need ongoing resourcing including financing and development of local supply chains and services. Efforts to promote behaviour change with information interventions in conditions of poverty must recognize the complex threatscapes that people contend with, and the limited potential for impact of fear appeals without concurrent increases in self-efficacy.

We note that this study, bounded as it is to focus on individual decision-making, omits four important systemic considerations. First, sharing monitoring information can influence cooperation between community, bureaucratic, and market-based stakeholders [34], so monitoring and reporting programme designs should account for this important context. Second, our qualitative findings reflected that perceptions of threat have personal and interpersonal dimensions, but our analysis did not explore this in-depth. Further work in this area could draw insight on collectivist and individualist orientations from cultural theory [91]. A third area for further study may be to explore how water managers respond to data on multiple water quality threats. In this study we focused on LWM responses to *E. coli* test results, but our work to understand water safety management in context also included chemical water quality monitoring. As a condition of government approval of our study, we did not report chemistry results to study participants but our survey and interview results indicate the importance of considering multiple water quality threats when engaging with water safety management options. Lastly, we have noted how the design of water supply infrastructure and management arrangements influences efficacy, but our sample size is too small to observe an effect of scale. The scale of supplies managed by LWMs varies from wells that serve a few households to piped distributions schemes that serve hundreds. Further research might usefully consider how scale influences LWM perceptions and behaviour in response to water quality monitoring.

## Figures and Tables

**Figure 1 ijerph-19-00597-f001:**
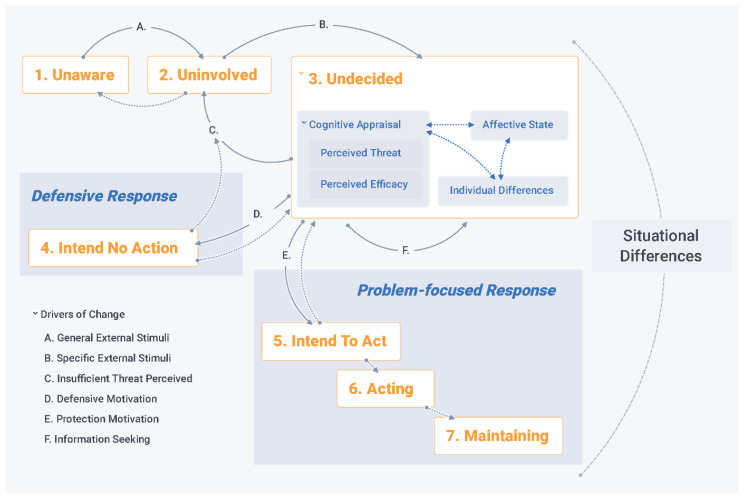
Integrated fear appeal framework situating the extended parallel process model (EPPM) within the precaution adoption process model (PAPM). The numbered stages of change (in orange) are drawn from the PAPM [54]. The message processing and outcomes concepts (in blue) are drawn from the EPPM [55]. The drivers of change are drawn from both models as explained in the text.

**Figure 2 ijerph-19-00597-f002:**
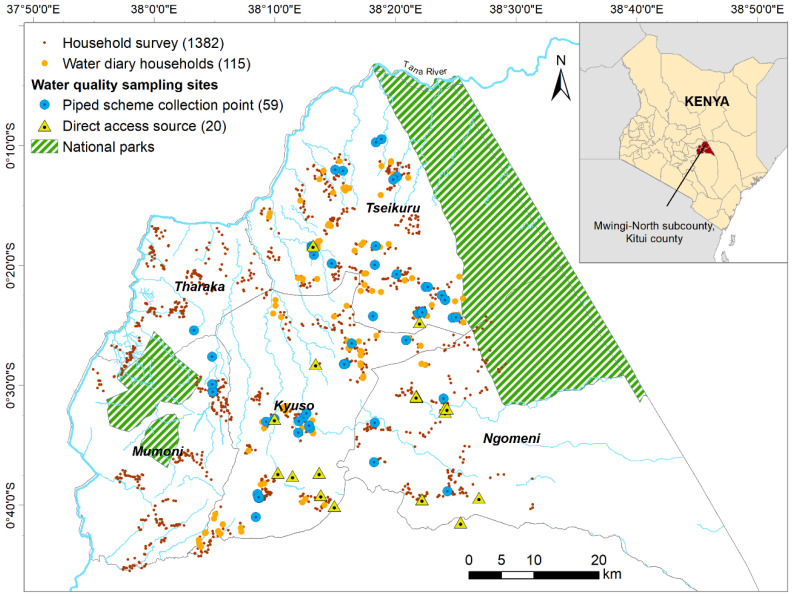
Map of the study area showing surveyed households and water quality monitoring sites.

**Figure 3 ijerph-19-00597-f003:**
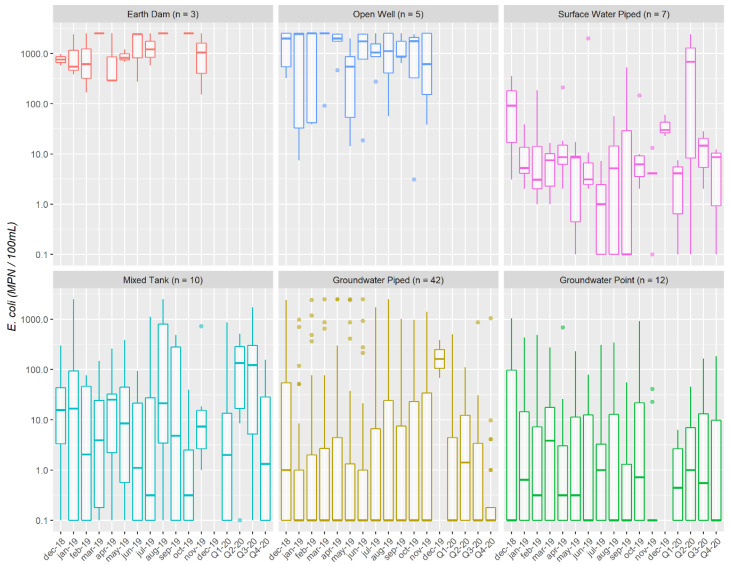
*E. coli* results for 79 water collection sites. Sites were monitored monthly from December 2018 through 2019 and quarterly in 2020. Those that were not registered for maintenance services (*n* = 34) were not monitored in 2020.

**Figure 4 ijerph-19-00597-f004:**
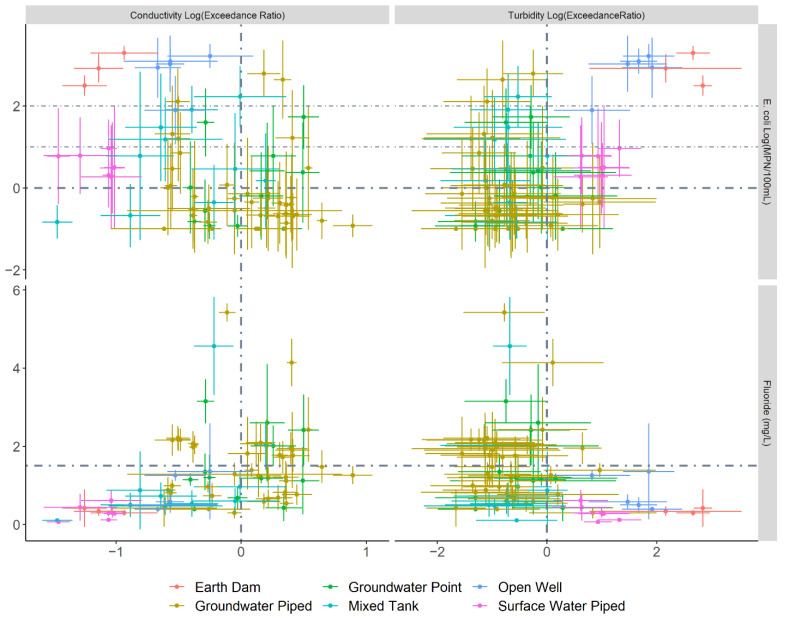
Water quality scatter plots comparing key threat parameters (*E. coli* and fluoride) with key organoleptic parameters (conductivity and turbidity) at 79 water collection sites. Dots show means and error bars show plus and minus one standard deviation from the mean. Log_10_ values are used for conductivity, turbidity, and *E. coli*. The conductivity and turbidity results are reported as exceedance ratios, which are calculated by dividing the test result by the East Africa Standards (referenced by the Kenya Bureau of Standards as KS EAS 12:2018) for conductivity (2500 µS/cm) and turbidity (25 NTU) in natural potable water. Since exceedance ratios are used, negative log values indicate results meeting the standard and positive log values indicate results exceeding the standard, as demarcated by the dot-dash lines. For *E. coli*, the dot-dash lines correspond to the WHO risk classification thresholds: log values below 0 correspond to low risk (<1 MPN/100 mL), log values between 0 and 1 correspond to intermediate risk (1–10 MPN/100 mL), log values between 1 and 2 correspond to high risk (11–100 MPN/100 mL), and log values above 2 correspond to very high risk (>100 MPN/100 mL). For fluoride, the dot-dash line corresponds to the WHO guideline and Kenyan standard of 1.5 mg/L.

**Figure 5 ijerph-19-00597-f005:**
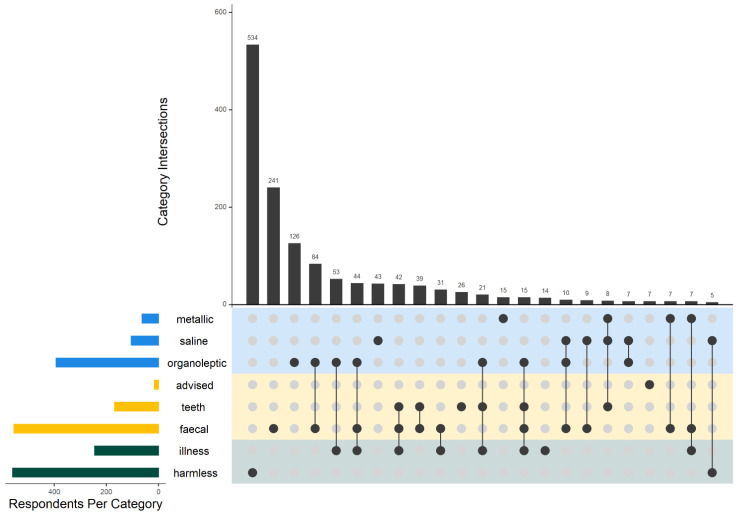
Intersecting sets visualization [75] showing the factors influencing household survey respondents’ judgements of drinking-water safety. The vertical bars show frequencies of judgement combinations in decreasing order; the horizontal bars on the left show the number of respondents that answered positively for each category. Blue corresponds to sense-based judgements including metallic taste, saline taste, or other organoleptic observations for taste, smell, and visual. Yellow corresponds to learning-based judgements including advice from others, knowledge about damage to teeth, or knowledge about faecal contamination hazards. Green corresponds to attribution-based judgements including whether respondents have attributed illness to drinking-water or not.

**Figure 6 ijerph-19-00597-f006:**
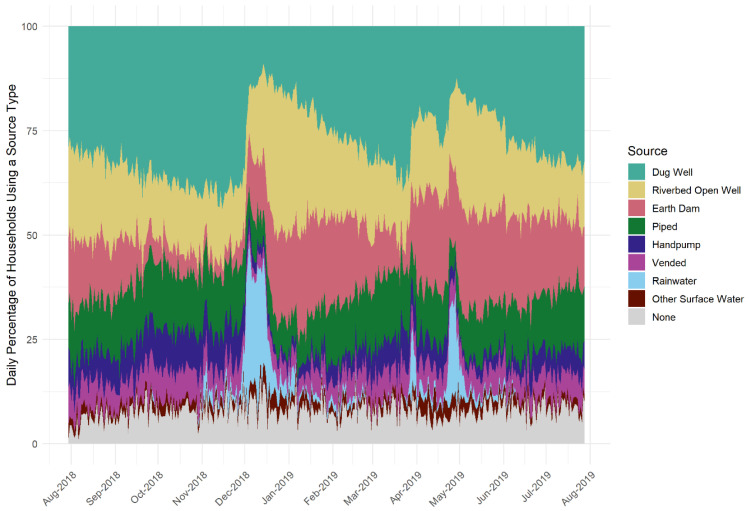
Percentage of water diaries households using a given water supply type for drinking and cooking over a year starting 30 July 2018.

**Figure 7 ijerph-19-00597-f007:**
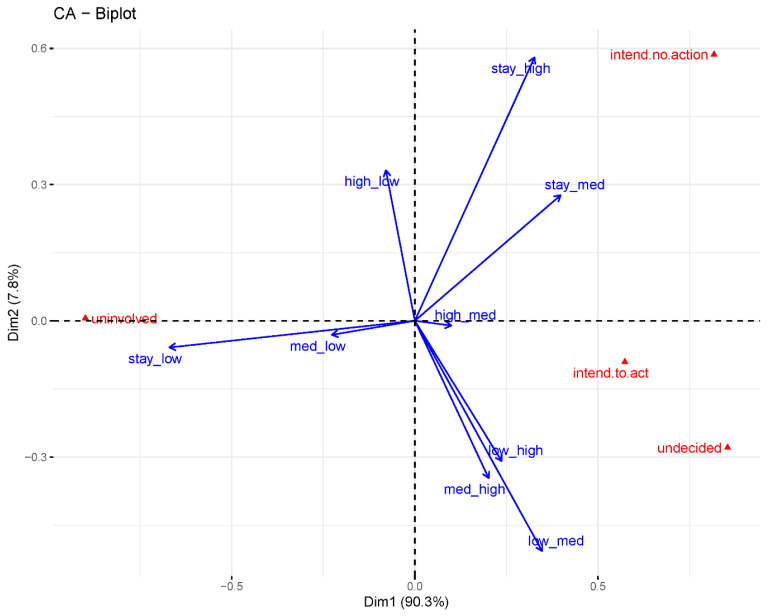
Contribution biplot of stage of change by change in perceived susceptibility. The dependent variable categories (red points) are positioned further from the centre of the chart if they contribute more strongly to the correspondence analysis solution (if they are more strongly associated with categories of the independent variable). Likewise, the independent variable categories are represented by blue arrows that are longer if they contribute more to the solution (if they are more predictive of the dependent variable). The angular distances between the arrows and the axes shows how much the independent variable categories contribute along each axis: the closer the arrow is to an axis, the stronger the contribution to that axis relative to the other one. If an arrow is midway between the two axes, it contributes to them equally.

**Figure 8 ijerph-19-00597-f008:**
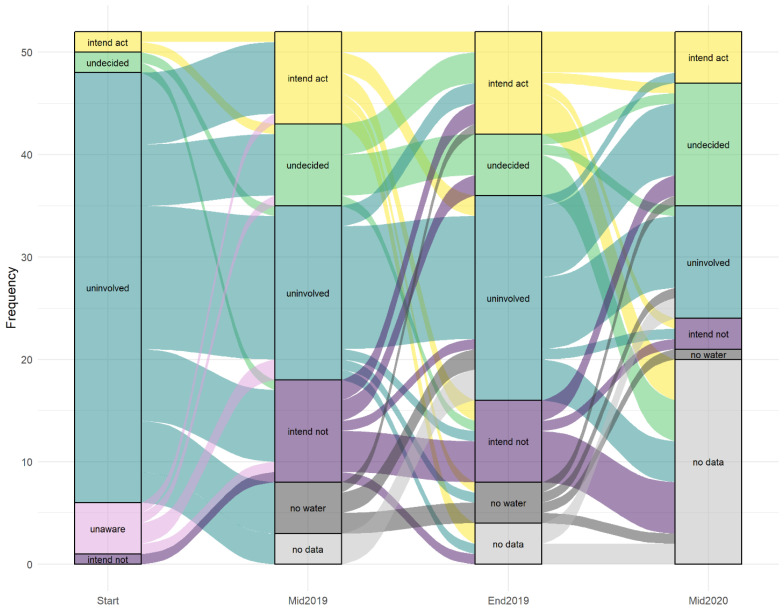
Evolution of LWM stage of change over the study period.

**Table 2 ijerph-19-00597-t002:** Key water user perception and practice themes from the interviews. Themes 1 to 4 relate to perceived threat; themes 5 to 8 relate to perceived efficacy; themes 9 to 12 relate to problem-focused and/or defensive responses.

Theme	Cases	% Coding Coverage ^1^	Description	Example Quote
(1) Surface water is especially unsafe	31	4.0 (1.3–8.3)	Participants pointed to the openness and stagnation of water as hazardous, and they linked the threat of disease (speaking of typhoid, amoebiasis, cholera, dysentery, stomach problems and diarrhoea) to inadequate separation of water from livestock, wildlife, latrines, and open defecation, with ‘dirt’ or ‘faeces’ carried into the water by rain (overland flow), on people’s shoes, or on containers and ropes that are used to draw water.	“There are places where people have not dug pit latrines, there are animals that have died and decayed in the bushes, and other bad things. When it rains, then all that dirt is swept by the rainwater and drained in the earth dam. Even now, the rain is not here but whatever dirt was brought before is still in the water source.”—P35F
(2) There is a microbial vs. chemical quality trade-off for groundwater	20	1.6 (0.5–4.2)	Participants recognized that groundwater is better protected from faecal contamination but they highlighted that the suitability of many groundwater supplies for drinking and cooking purposes is limited by salinity and bitterness, especially during dry seasons. Participants linked salty water to unquenched thirst, constipation, bloating, and gastrointestinal pain, which one woman described as “slashing your intestines into pieces”. Participants asked about the potential health impacts of salinity on livestock, but they did not discuss chronic health consequences for themselves.	“Water from the boreholes is safe for human consumption since it is well covered and protected from all sources of contamination. However, … it is limited in use due to its saltiness.”—P07M
(3) Lack of specific external stimuli limits judgement of water safety	19	1.7 (0.2–5.4)	General knowledge about water contamination is widespread, but none of the participants had received test results for the water supplies that they relied on. Participants discussed the limitations of assessing water quality based on organoleptic properties. On the one hand they may have a bad reaction from drinking water even if it appears clean but, on the other hand, when they become sick they usually cannot be confident of the cause.	“You have to realize that even if the water is dirty, we cannot tell because we don’t have a professional to check its quality or treat it. We just take the water the way it is, even when you get sick you can never tell whether it was the water or something else.”—P06M
(4) Water quality threats induce fear for oneself and others	25	1.8 (0.2–3.9)	Participants spoke of prolonged stomach pain and needing to seek medical relief. Death and the contribution of waterborne illness to malnutrition were not directly discussed, but participants said that they fear dirty water and that infants are more susceptible to hygiene-related illness, including from unclean water. This view of heightened susceptibility extended to adults who are already weakened from illness.	“We have a lot of fears because, personally, I have stomach problems and if I take the water without boiling then the problem escalates. I also fear for my children because some of them have similar stomach problems.”—P04F
(5) Despite knowledge of threats, poverty constrains safe water practices	20	3.5 (0.3–12.0)	Participants differentiated know-how, will, and capability to act. They discussed access and affordability issues that prevent them from acting on knowledge about water safety practices. They also highlighted the inability of communities to maintain NGO projects without ongoing support, especially in the face of difficult environmental conditions, vandalism, and theft.	“We were trained about the earth dam water and told that it is not clean, but due to our low-income levels and other problems we have here you may find people drinking the earth dam water just the way it is knowing very well it is not good for drinking.”—P35F
(6) Gender norms especially limit the self-efficacy of women	31	4.8 (0.7–13.7)	Gender norms within families and the wider community limit opportunities for women to lead and participate in water management committees. Further, many water supplies have flexible payment structures that require users to strike an agreement with the owner or management committee. In most cases, the household head (usually men) makes these agreements, they also decide what portion of household income can be spent on water; consequently, they largely determine supply selection even if other household members (usually women) fetch water and manage its use within the household.	“I cannot say I have anything I do for livelihood, maybe a business or anything. I like the idea and I would very much want to do that, but my husband refuses... And this happens for most women. This really affects us in terms of provision for our children... you will find that [I] am the most knowledgeable person about the needs of the children and the household... Even when they are aware that we know all these, they say it is not possible to allow us to go sell their produce.”—P04F
(7) Water source protection is a collective action challenge	25	2.4 (0.4–6.6)	Participants emphasized that self-efficacy is eclipsed by the need for collaboration and leadership from committees or owners in protecting water sources. They discussed examples where protective measures have failed due to lack of cooperation, presenting them as testament to the difficulty of sustaining protective measures despite strong motivation—water quality is only part of the motivation, participants were also concerned about drowning accidents, water shortages, and functionality issues.	“… it is very dirty, people have allowed [livestock] to enter the earth dam and urinate among other things... The thing is if you go and complain, no one listens to you. So, after a while you stop worrying and do what others are doing. If the consequences come, they affect you all.”—P06M
(8) Rural isolation limits self-efficacy	19	2.0 (0.4–5.3)	Participants noted the lack of follow through on campaign promises and expressed a sense of isolation both by physical distance and political hierarchy. None of them were positive about their ability to attract or mobilize support from NGOs or the government (neither through the former system of chiefs nor the post-devolution system of village administrators).	“I think we are very deep in the rural areas, I don’t even know how you’ve reached here (chuckles), because nothing ever gets here. People only get to this area when they are in need of votes.”—P29F
(9) Supply selection is influenced by multiple dynamic factors	35	11.0 (4.0–28.8)	Water supply selection varies in response to rainfall, distance, queuing, security, labour, monetary cost, livestock needs, personal relationships, functionality, and quality. Groundwater salinity limits alternatives to unprotected water supplies and is, therefore, a key constraint on collecting safer water. Further, distance and cost are even firmer constraints on choice than preference of different water qualities. Payment structure is also important: where people can borrow, pay with food, or offset monetary payments by providing labour for the maintenance of a water point, they can more consistently access a preferred supply. Supplies that require upfront payment in cash without exception are more challenging.	“When it gets very dry, the water gets saltier, but when it rains well, the salt is reduced—though not all the times... [In the dry season], people have to buy fresh water from the market kiosks which amounts to being very expensive for some of the community members... Unless one buys water from the salt-less wells, which are very few like three wells in this area.”—P11F
(10) Problem-focused water safety measures are employed intermittently	20	1.5 (0.3–3.0)	Boiling, adding chlorine disinfectant, filtering water, or buying bottled water is done intermittently in response to specific stimuli including advice from doctors, to provide for new infants, or to protect people who are already ill. The key reason for not consistently maintaining measures to protect against water quality threats is that time, energy, and money must be put towards problem-focused responses to many different threats, some of which are more immediately severe than waterborne diseases.	“When a person fetches water and takes it home, most of them use it without doing anything to it not even treating it or even boiling it; but when they are told they have amoeba or typhoid, they start boiling the water or even use Water-Guard to treat the water.”—P13M
(11) Resources must be balanced for problemfocused responses to multiple threats	29	2.7 (0.3–7.3)	In adopting problem-focused behaviour, participants balanced water quality threats against many others including attacks from people, hyenas, snakes, and *majini* (spirits) when walking to fetch water; thirst and fatigue from inadequate water access; unreliable rainfall and crops; flooding; falls into wells or reservoirs; chest problems from cooking fires; and communicable diseases including HIV. Participants also worried about keeping children in school. For girls, this intersected with concerns about gender-based disempowerment, sexual assault, early pregnancy, and abusive marriages. For boys, it intersected concerns about drug and alcohol abuse.	“People are having problems finding money to buy water. At the same time, they are also scared of selling their food to leave the children with nothing to eat. The fear is also because no one is sure that it will even rain.”—P30F
(12) Cognitive reappraisal, particularly resignation, is a common defensive response	25	1.9 (0.3–6.4)	Participants framed their circumstances as uncontrollable; they were resigned to “use patience” and “persevere with the situation at hand”. One participant linked feeling a heavy burden to using resignation to “try navigate the challenges”. Other forms of cognitive re-appraisal were also expressed including religiosity (circumstances are in God’s hands), downward comparison (unsafe water is better than no water), self-exemption (the hazard is real, but I am not susceptible), and humour as reframing.	“For lack of alternative a woman can even start having labour pains when she is on her way from the water point... Some even suffer backaches up to now. But then how can we help them? This is how the world is.”—P20F

^1^ Coding coverage is expressed as: average (min–max).

**Table 3 ijerph-19-00597-t003:** Patterns of LWM response to monitoring results reports.

No.	Description	Main Cases	Secondary Cases	Intentions ^1^	Actions ^1^	Efficacy Gap ^2^	Defensive Responses ^1^
1	Reporting does not motivate formation of intentions. This pattern is associated with positive affect displays in response to sustained low perceived susceptibility (consistent absence of *E. coli* when the water was in regular use).	7	0	0 (0–0)	0 (0–0)	0 (0–0)	0 (0–2)
2	Reporting prompts intention to act proactively (based on potential future threat but not current threat) or intentions to act are extended to other supplies. Despite *E. coli* negative results, accompanying information and/or prior test results prompt intention to act and information seeking (specifically, requesting test results from nearby alternative supplies for comparison). This pattern is associated with positive affect displays in response to low perceived susceptibility, but it is unclear what separates uninvolved LWMs (pattern 1) from those who extrapolate threat information and are motivated to engage with water safety measures despite *E. coli*-negative test results.	7	10	2 (1–3)	1 (0–2)	1 (0–2)	0 (0–2)
3	Reporting of a reduction in *E. coli* concentration relative to a previous test result prompts reversion from intention to act to uninvolved. This pattern is associated with positive affect displays in response to reduced perceived susceptibility relative to a previous report.	8	14	2.5 (0–3)	1 (0–2)	1 (0–3)	0 (0–2)
4	Sustained threat prompts initial variable engagement that evolves to uninvolved. This pattern is associated with regular concentrated *E. coli* contamination and the highest expected efficacy gaps, meaning the difference between the number of intentions voiced and the number of actions taken was largest for these LWMs. Defensive processing was most common for these LWMs, presumably supporting their return to the uninvolved stage, wherein they no longer consider adopting control measures.	7	0	4 (1–5)	1 (0–2)	2 (1–4)	3 (2–4)
5	Variable threat prompts long-term engagement with LWMs moving between indecision, intentions to act, and intentions to not act. This pattern is associated with changing perceptions of susceptibility and partial efficacy (actions do not fully control the threat). Defensive processing is indicated but respondents continue to acknowledge and want to address the threat. Respondents looked to test results for confirmation of the impact of their actions.	16	5	3.5 (1–5)	3 (1–4)	1 (0–1)	2 (0–4)
6	Sustained threat prompts sustained intention to act. This pattern is associated with regular concentrated *E. coli* contamination and is differentiated by higher self-efficacy with LWMs having know-how and ongoing resourcing to implement consistent chlorine disinfection (with support from NGOs in four out of the seven cases).	7	1	3 (2–8)	2 (2–5)	1 (0–4)	1 (0–3)

^1^ Values are medians (with range in brackets) of the number of intentions, actions, and defensive responses recorded at least once for each LWM. ^2^ Values are medians (with range in brackets) of the expected efficacy gap for each LWM, calculated as the number of actions subtracted from the number of intentions.

## Data Availability

The cross-sectional household survey data presented in this study are openly available in the UK Data Service ReShare repository at 10.5255/UKDA-SN-854561. The longitudinal household survey data and the water quality monitoring data are available on request from the corresponding author. The interview transcripts are not publicly available due to confidentiality restrictions.

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
