# Peer review of "Fear, Efficacy, and Environmental Health Risk Reporting: Complex Responses to Water Quality Test Results in Low-Income Communities"

_ijerph, 2022, doi:10.3390/ijerph19010597_

Round 1

Reviewer 1 Report

General comments:

The paper presents a really interesting analysis of results of water quality reporting programmes in rural Kenya. However, I think considerable work is required to make the paper suitable for publication.

Firstly, the authors have neglected some key literature which I think is of relevance. In particular key literature around management arrangements (including some recent work on ‘going with the grain’) and operation and maintenance of water supplies is missing.

The authors make only very limited reference to the water quality results outside the water quality results section. There is only very limited reference to the water quality results in the context of the other results (threat and efficacy, water management, susceptibility, etc). While the authors do caveat the fluoride, salinity and turbidity results as important primarily for context, it is a bit disappointing not to see these used elsewhere in the analysis. I think the link between water quality (including the E-coli results) and the other aspects of the analysis need to be strengthened particularly in the discussion, where at present there is not reference to the water quality results at all.

Why not break up the discussion to aid the reader by using section headings to discuss key themes? Perhaps the structure of the discussion could be more clearly related to the structure of the results section?

Finally, given the effort the authors went to develop the ‘integrated fear appeal conceptual framework’ it is surprising how little it is used within the discussion to frame and contextualise your results. Can you more clearly use the conceptual framework to discuss your results? Perhaps this may also help bring the water quality results into the discussion.

I think some wider context is also required. There is limited reference to other areas where risk reporting has been tried, for example it would have been interesting to at least make the reader aware of the red and green handpumps approach to arsenic risk in Bangladesh. Are there lessons that can be learned from experiences elsewhere?

Specific comments:

Line 57: It would be good to slightly widen the literature you cite on community level management here.

Line 68: See MacAllister et al. (2020) for an example of where monitoring served near-term operational decision making effectively during drought in Ethiopia (https://www.nature.com/articles/s41467-020-14839-3). It’s an example of infrastructure monitoring rather than water quality monitoring but I still think holds relevance here.

Line 195 – 211: Could you provide a map of the study area, areas where you sampled water quality and conducted household surveys and interviews?

Line 215: Hyperlink is not working.

Line 238 – 239: What types of water supplies were included in your study? Can you list these here as you do for management models (but not necessarily to the same level of detail as in table 1)?

Line 241 – 242: I would be careful with terminology here; mechanised supplies may imply to many readers larger systems such as diesel/solar systems with submersible pumps and possibly even piped distribution. You also examined systems including handpumps on boreholes or protected wells and think this needs to be clearer.

Line 243: Can you be clear what you mean by fit-for-purpose laboratory here? There are no details of the laboratory in instrument 4 in Table 1, only details of field parameters measured and instrumentation used. In Table 1 instrument 4 you also refer to field lab analysis? What does that mean? It appears you only conducted field measurement of water quality parameters, if this is the case, it would be better to make that clear. If only field measurements were made the use of the terms ‘lab’ or ‘laboratory’ is confusing.  Did you conduct any QA/QC by sending duplicates to labs, particularly for fluoride? Why were the parameters in Table 1 instrument 4 selected, it is not clear? Was it based on previous studies in the area, consultation with managers/communities or a combination?

Line 309 – 310: Not clear what you mean by ‘key trade-offs’ here. Can you explain this in more detail?

Figure 2: There is a lot of detail in this figure at present and it could be quite hard for a reader to follow. Is it possible to present this in a different way that simplifies things? Could you perhaps separate by water source type? Alternatively, could you provide an additional figure, perhaps as a box-plots with actual measured values to summarise the water quality results before presenting the more detailed analysis presented in this figure? The details in this figure may also require a bit more explanation in the text, see also my comment about lines 3019 – 310 above.  

Line 340 – 342: Possibly in the discussion, but might be good to provide some context to these results here. Were these eight villages in areas with known fluoride problems?

Line 379: Hyperlink is not working.

Line 398: Hyperlink duplicated.

Figure 4: Really interesting figure, nicely presented.

Line 498 – 531: Figure 5 is duplicated and text is strangely formatted.

Figure 5 (version 1): I would suggest discussing the associations listed here in context within the main text rather than in the figure.

Line 543 – 544: What do you mean by ‘water supply design’ here? Do you mean water source type? What do you mean by ‘starting condition’? Needs to be clearer.

Figure 6: I think the most striking thing here is the increase in ‘undecided’ proportion of response by mid-2020.  Might this have been driven by other external factors (e.g. Covid?!). Prior to this there did seem to be small increases in ‘intend to act’ and decreases in ‘undecided’ and ‘intend not’ proportion of responses. None of this is really discussed in the text.

Line 558 – 559: Important finding, which is consistent with previous literature on community management and functionality. I think it would be good to bring this wider literature into the discussion here.

Line 649 – 650: Again, its not clear what you mean by ‘water system design’? The confusion arises I think because it could appear to the reader that you are using the terms ‘water source type’, ‘water supply type’ and ‘water system design’ interchangeably. This needs clarity because ‘water system design’ may include protective measures, such as a well-constructed platform and fencing. ‘Water source type’ really only refers to the resource and the lifting/extraction technology, ‘water supply type’ is potentially slightly broader but may not include variations in details of the design (for example source protection measures). I suggest using consistent terminology throughout.

Line 650 – 652: There are a wide array of studies that show the need for external support for on-going operation and maintenance (including one example provided earlier in my comments) and I would like to see, at least some of, these referenced here.

Line 670 – 684: While this paragraph is very interesting, it was quite hard to understand how it relates specifically to your results. Can you make the connections much clearer and contextualise this part of the discussion better? Otherwise it feels a bit superfluous.

Line 699 – 701: Very interesting and important finding.

Line 711 – 713: What would be the mechanism for communities to receive these results if requested? Would this not place a burden on already stretched and under resourced management arrangements (whether they are community based, government, private, etc)?

Line 717 – 722: The references here to willingness to pay comes a bit out of the blue. I didn’t see any mention of it in your results section. Its not clear to me how your results are relevant to questions around willingness to pay. Can you make the link clearer? Otherwise I suggest remove.

Author Response

Dear reviewer 1,

Many thanks for the time and consideration that you put into reviewing our paper. We greatly appreciate your thoughtful comments and feel that your input has helped us to improve the paper. We have attached a PDF with our detailed responses to each of your comments. We also included out responses to the other reviewers so that you can see the reasons behind additional changes in the revised manuscript. Thank you again for your time in engaging with our work.

Reviewer 2 Report

Problems of drinking water pollution, compliance with designated water quality standards, and proper management of water resources are significant for the functioning of the economy and society. These problems do not concern only rural areas and low-income countries, although they are more common there. Tools that enable control and possible quick corrective actions in this respect seem to be necessary. Therefore, I believe that the manuscript's subject matter is essential, timely, and may interest potential readers. The structure of the manuscript is correct; however, I have several comments for the authors.
In the "Introduction" section, the authors have focused mainly on presenting water resources management problems in Africa. To raise the profile of the topic, I suggest mentioning that similar research is being conducted around the world, for example, in Europe. You can refer to the studies: https://doi.org/10.3390/en14133841 and http://dx.doi.org/10.1515/mspe-2019-0030
At the end of the "Introduction" section, please briefly outline the different structures of the manuscript, and it will improve readability and clarity.
In the Conclusions section, please clearly and separately present how the research can help water managers and help water consumers.
I congratulate the authors for taking up an essential and exciting issue, and I believe that after the corrections, the manuscript will be published.

Author Response

Dear reviewer 2, 

Many thanks for the time and consideration that you put into reviewing our paper. We have attached a PDF with our detailed responses to each of your comments. We also included our responses to the other reviewers so that you can see the reasons behind additional changes in the revised manuscript. Thank you again for your time in engaging with our work and for helping us to improve our paper.

Reviewer 3 Report

The issue under investigation is topical and important as it discusses an area of water studies that is often underlooked. Water quality and health risk communication are important areas that require more investigation. While this was an interesting article to read, the sentence construction (language) made it difficult to understand what the scholars want to say. See the long sentence (lines 48-51, 90). In the introduction, the authors start discussing the materials and methods as well as the research findings. Shouldn't they be in their appropriate sections? The authors make use of and integrate two relevant and informative frameworks for the study. They also make use of a comprehensive research design and methodology. When presenting the results, should the description of the Figure form part of the citation? For example, Figure 2? The study yielded good and interesting results but I feel there could have been two papers or more from this one paper. The authors present significant and overwhelming data. I believe this is because the paper is informed by a research project. Focus on one issue, for example, the evaluation of risk reporting alone would have made a clearer and more interesting contribution to risk communication literature. And also, using just one data collection method like interviews.   

Author Response

Dear reviewer 3,

Many thanks for the time and consideration that you put into reviewing our paper. We have attached a PDF with our detailed responses to each of your comments. We also included our responses to the other reviewers so that you can see the reasons behind additional changes in the revised manuscript. Thank you again for your time in engaging with our work and for helping us to improve our paper.

Reviewer 4 Report

Thank you for a detailed and well-described study.

The study aimed to assess long term (1 and half years) behavioural changes based on drinking water quality risk communication through qualitative methods (cross-sectional and longitudinal surveys, interviews and water quality tests) in the rural community of Kitui, Kenya with a particular focus on water users and Lay Water Managers (LWMs).

Some minor typographical edits are attached in the review comments for your consideration.

Also, I suggest that the manuscript should be made more concise across sections though the scale of the study is understandable to warrant it. 

Author Response

Dear reviewer 4,

Many thanks for the time and consideration that you put into reviewing our paper. We have attached a PDF with our detailed responses to each of your comments. We also included our responses to the other reviewers so that you can see the reasons behind additional changes in the revised manuscript. Thank you again for your time in engaging with our work and for helping us to improve our paper.

Round 2

Reviewer 1 Report

I thank the authors for their detailed responses to all of my comments and also to those of the other reviewers. The revised manuscript is much improved and I think will make a valuable contribution to the literature. I have some additional minor comments below.

One thing that I think remains largely unaddressed is the consequences of non-reporting of water chemistry results. I appreciate that you are restricted by the conditions of the research but it feels to me as if this could have an important bearing on your results. Can present some hypothesis as to how this might affect things, possibly using participant responses in the diaries? How might non-reporting of water chemistry affect people’s responses when they are aware of other water quality problems? For example, users expressed concerns about groundwater salinity in certain areas, and it would seem might rank this as a more significant issue than e-coli.

Non-reporting of water chemistry results might be a particular issue when communities do not experience a high level of disease burden from (e-coli) contaminated water. From the text we have only limited feeling of the level of disease burden in these communities. You report DALYs but what about actual rates/levels of occurrence of diarrheal disease? Knowing this might be useful context and it may help explain some of your results in terms of engagement (e.g. Figure 8).

I was very interested to read that in areas where there was an awareness of the link between dental problems and water quality, the water quality results actually showed that fluoride levels were not elevated relative to the other samples. Again, how might existing perceptions affect people’s willingness to engage with what is presented to them, i.e. only a partial picture of the water quality situation? I understand the limitations the study faced but I would be interesting to see some reflection on these issues.

I think the use of water supply design and starting condition terminology in the context you have used it is still rather confusing. What about using systems terminology throughout the manuscript (i.e. something like ‘integrated water supply system’)? This would also avoid the use of the term starting condition, which I don’t think is helpful because the starting conditions don’t change throughout your study (it’s not like we are setting up a model with initial conditions, conditions which then change through the model time-steps). When using whatever systems terminology you choose just be clear that this includes infrastructure and managements arrangements. I would also state explicitly, perhaps in lines 259 – 261 in the revised manuscript, that ‘source type’ refers to groundwater, surface water or rainwater and that ‘supply type’ refers to the lifting/extracting and distribution set-up.

Author Response

Dear reviewer 1, 

Thank you again for your time in reviewing our work. Please find attached our responses to your comments for minor revision. As before, our line number references correspond to the manuscript with tracked changes.

Best wishes, 

the authors

Reviewer 2 Report

 A revised version of the manuscript is suitable for publication

Author Response

Reviewer 2, 

Thank you for approving our revised manuscript for publication and for all of the time that you spent on reviewing our work. 

Best wishes,

the Authors